# Implementation Factors for the Delivery of Alcohol and Drug Interventions to Children in Care: Qualitative Findings from the SOLID Feasibility Trial

**DOI:** 10.3390/ijerph18073461

**Published:** 2021-03-26

**Authors:** Hayley Alderson, Ruth McGovern, Alex Copello, Elaine McColl, Eileen Kaner, Deborah Smart, Paul McArdle, Raghu Lingam

**Affiliations:** 1Population Health Sciences Institute, Newcastle University, Newcastle NE2 4AX, UK; r.mcgovern@newcastle.ac.uk (R.M.); elaine.mccoll@newcastle.ac.uk (E.M.); Eileen.kaner@newcastle.ac.uk (E.K.); Deborah.Smart@newcastle.ac.uk (D.S.); 2School of Psychology, The University of Birmingham, Birmingham B15 2TT, UK; a.g.copello@birmingham.ac.uk; 3Research Birmingham and Solihull Mental Health Foundation Trust, Birmingham B1 3RB, UK; 4CAMHS, Northgate Hospital, Morpeth, Northumberland NE61 3BP, UK; Paul.McArdle@cntw.nhs.uk; 5Population Child Health Research Group, School of Women and Children’s Health, Faculty of Medicine, University New South Wales, The Bright Alliance, High St & Avoca Street, Randwick, NSW 2031, Australia; r.lingam@unsw.edu.au

**Keywords:** alcohol, drugs, psychosocial interventions, qualitative, children in care

## Abstract

Young people in care have a four-fold increased risk of drug and alcohol use compared to their peers. The SOLID study aimed to deliver two behaviour change interventions to reduce risky substance use (illicit drugs and alcohol) and improve mental health in young people in care. The study was carried out in 6 local authorities in the North East of England. Young people in care aged 12–20 years, who self-reported substance use within the previous 12 months were randomised to Motivational Enhancement Therapy, Social Behaviour and Network Therapy or control. In-depth 1:1 interviews and focus groups were used with young people in care, foster carers, residential workers, social workers and drug and alcohol practitioners to explore the key lessons from implementing the interventions. The Consolidated Framework of Implementation Research framed the analysis. Findings illustrated that the everyday interaction between individuals, service level dynamics and external policy related factors influenced the implementation of these new interventions at scale. We concluded that unless interventions are delivered in a way that can accommodate the often-complex lives of young people in care and align with the drug and alcohol practitioners’ and social workers priorities, it is unlikely to be successfully implemented and become part of routine practice.

## 1. Introduction

One in every 156 children in the UK is ‘looked after’ by a local authority, alternatively known as being in care [1]. Many young people in care have been exposed to the co-occurrence of parental substance misuse, parental mental illness and domestic violence and inevitably will have experienced adverse childhood experiences [2,3] further increasing their risk of personal substance use and poor life chances [4]. In the UK, 42% of young people living in a residential children’s home and 25% of young people in foster care had drunk alcohol at least once a month, compared to 9% of young people not looked after [5]. Similarly, a UK national survey of care leavers (young people transitioning out of local authority care) showed that 32% smoked marijuana daily [6] compared to an estimated 5% of young people aged 16–24 years in the general population [7], with 11.3% of looked after children aged 16–19 years having an identified substance misuse problem [8]. The Chief Medical Officer and the National Institute for Health and Care Excellence [9] highlighted the urgent need for effective interventions to reduce risky drug and alcohol use in this vulnerable group of young people [10]. If young people in care are to reach their full potential, it is essential that they receive more ‘targeted support’ at the right time, given their needs may be greater in comparison to their peers who are not in care. Two evidence-based interventions shown to be effective in decreasing levels of substance misuse in adolescents and young adults more generally [11] are Motivational Enhancement Therapy (MET) [12] and Social Behaviour and Network Therapy (SBNT) [13]. Both are psychologically focused interventions which aim to enhance motivations and skills for positive behaviour change by working through internal mechanisms, with or without externally driven considerations [14].

MET is a client centred, directive counselling approach, developed within the NIH MATCH study as a concentrated version of motivational interviewing, which adds a problem feedback component to standard treatment [12,15]. Miller et al. (1999) argue that the distinctive assumption of MET is that the responsibility for change and the motivation to elicit change lie within the client. When delivering the MET intervention, it is the practitioner’s role to develop the clients’ motivation by mobilising their inner resources and helping the client change their thinking to facilitate behaviour change [12]. Carney and Myers’ (2012) systematic review concluded that Motivational Interviewing and MET have shown therapeutic promise for adolescents with problem substance use [16,17,18].

In contrast, SBNT is a systematic psychosocial approach, that stresses the client’s social interactions and social support and utilises cognitive and behavioural strategies to help clients build social networks supportive of positive behaviour change in relation to problem substance misuse and goal attainment [19]. As an approach, SBNT, recognises that drug and alcohol use is not an isolated phenomenon but rather occurs in the context of the person’s social world that affects and is affected by others e.g., family members, carers, friends. It is known that clients have an increased chance of successful behaviour change if they can draw support from networks available to them; it is the practitioner’s role to help galvanise a support network for their client [19]. The approach of developing a team around the young person is well established and supported by the National Institute for Health and Care Excellence [9].

The SOLID (Supporting Looked after children In Decreasing Drugs, and alcohol) pilot randomised controlled trial aimed to evaluate the feasibility and acceptability of a definitive effectiveness trial of these two behaviour change interventions MET [12] and SBNT [19] compared to care as usual (control) to decrease the risk of substance use and improve the mental health of young people in care. The study focused on secondary prevention and attempted to manualise a care pathway within the current social care system.

A detailed protocol paper outlining the different phases of the pilot RCT [20], a paper describing the process of adapting and manualising the interventions delivered within the study [21] and a paper describing the process and results of the pilot RCT [22] have been published, a consort diagram is shown in Figure A1. In summary, 860 participants were screened, 211 (24.5%) met the inclusion criteria for the trial. Of the 211 eligible young people, 112 (53%) were recruited and randomized and just 15 of the 76 (20%) participants allocated to the adapted interventions attended any of the MET or SBNT sessions [22,23] Whilst it was not feasible to progress to a full trial, several key lessons have been learned to inform research for young people in care.

As part of the feasibility study, a detailed, theoretically framed process evaluation was undertaken. This evaluation aimed to consider acceptability and engagement with the interventions and the trial procedures and to undertake a qualitative assessment of the barriers to successful implementation of the interventions by local drug and alcohol workers within existing health and social care pathways. The purpose of this paper is to present the qualitative implementation evaluation, inclusive of findings from young people offered the interventions, and practitioners involved in delivering the intervention and supporting the young people.

## 2. Materials and Methods

The qualitative process evaluation included semi structured interviews and focus groups. Semi structured interviews were chosen as the data collection method with young people due to the sensitive nature of the information they would be sharing regarding their substance use and associated risky behaviour. A mixture of interviews and focus groups were used with social workers, drug and alcohol practitioners and carers (foster carers and residential workers) to accommodate their preference and in a number of circumstances it was requested that the researcher attended a team meeting which facilitated access to multiple professionals at one time. The semi structured interviews and focus groups subsequent analysis were framed using the Consolidated Framework of Implementation Research (CFIR). The CFIR comprises five major domains, incorporating 39 constructs, relevant to the implementation of a novel intervention or policy. Damschroder (et al. 2009) have published a detailed description of each of the constructs, with a rationale and definition of each one [24]. The CFIR is summarised in Figure 1.

The study took place in the North East of England across six local authority sites. Each local authority site had a commissioned young people’s drug and alcohol service, that signed up to participate in the study and agreed that drug and alcohol practitioners within these services were trained to deliver either the SBNT or MET interventions as part of the intervention study arms. All the participating drug and alcohol practitioners were experienced in delivering psychological and social interventions to young people, though in many instances their previous training was not standardised. Practitioners had varied prior training in social work, youth work and counselling. Fourteen drug and alcohol practitioners working within the young people’s drug and alcohol services attended a two-day training course in either SBNT or MET delivered by members of the team experienced in using the approaches and delivering training. All practitioners received the relevant detailed manual and associated handouts to support the delivery of either the SBNT or MET intervention. Post training, the clinical trainers offered monthly supervision sessions to practitioners delivering the SBNT or MET interventions to support the implementation of the interventions in practice [23]. In addition, practitioners were requested to audio record delivery of the SBNT and MET sessions to assess the internal validity of the interventions. Unfortunately, due to the small number of available audio recordings, it was not possible for us to accurately assess the fidelity of the intervention sessions being delivered. This is reviewed in detail in our accompanying publication [22].

### 2.1. Participants

This paper presents data collected from 106 participants inclusive of: young people in care (*n* = 37) and professionals (*n* = 69). Process evaluation data collection took place between July 2017–November 2018 [23], participants took part in a one-to-one interview or focus groups as outlined in Table 1 below.

Informed written consent was obtained prior to each interview or focus group taking place. Assent was obtained from young people under 16 years, and consent from the individual with parental responsibility. Interviews took place within the practitioner’s workplace and for young people they took place within their home environment. Focus groups with foster carers and residential workers took place within the residential setting or a community venue. Prior to every interview, researchers ensured that participants had read and understood the participant information leaflet, reiterated that participation was voluntary and provided an opportunity for any questions to be asked and answered. Interviews were audio recorded and transcribed verbatim, pseudonyms were used within the transcripts to ensure that anonymity was upheld. The study was granted ethical approval by Newcastle and North Tyneside 1 NRES Committee (16/NE/0123) on 22 April 2016.

### 2.2. Analysis

The CFIR constructs served as a guide for our analysis and interpretation of qualitative data and provided a framework to report the implementation related findings. The key lessons learned from implementing the SOLID study in relation to the delivery of the adapted interventions within the trial processes and the evaluation of factors that impacted on delivering the interventions within a usual care pathway are reported in this paper. The paper reports findings that are about the implementation of the intervention and findings that are directly about the interventions themselves. To aid differentiation, the categorisation of the findings are identified by [implementation] and [intervention]. The main body of the results are structured using the CFIR constructs [24].

## 3. Results

Interview duration ranged from 20 to 40 min and focus groups were 60 to 90 min in duration.

### 3.1. Characteristics of the Interventions

The data relevant to this construct has been published previously [21].

### 3.2. Inner Setting—Organisational Characteristics, Culture and Readiness to Change

The inner setting comprises features of the implementing organisation and the individuals working within it which might influence the implementation effort. Drug and alcohol practitioners sometimes questioned the appropriateness of the study referrals in relation to their perception of the young people’s needs. The criteria for participants to receive an intervention was that the young person had reported substance use within the previous 12 months. Social workers and carers noted that practitioners often perceived young people to have needs below what they considered to be usual referral thresholds and projected a view that alcohol and drug use is a normalised part of adolescence. Despite recognition that looked after children were vulnerable to developing problematic levels of substance use, practitioners tended to take a treatment perspective to substance use (i.e., is it a problem now?) rather than seeking to intervene early to divert trajectories which may evolve into more problematic patterns of use:


*“It kind of all brings it back to asking those appropriate questions and getting that person the appropriate support. So, I mean, from where he came from, he’s actually in quite a good place to where he was. I said that to his social worker because we question him coming through full stop”*
(Karl, SBNT drug and alcohol practitioner [implementation]).

Interestingly, this young person was already known to Karl and this is what influenced the decision to say the referral was inappropriate despite the young person reporting use of alcohol and cannabis and being identified by a validated screening tool as being at risk. There appeared to be a level of desensitisation regarding what constitutes an ‘appropriate’ level of need to receive specialist support which may have influenced the resources allocated to young people.

The structural characteristics of the drug and alcohol services dictate that a treatment focused approach is taken and that only individuals with a dependency or significant ‘problematic use’ are eligible. Many young people in care did not consider their substance use to be problematic, and did not identify themselves as requiring support, therefore declining the offer of support from a specialist drug and alcohol worker:


*“I got offered [sessions] if I wanted the drug support or anything. I don’t need any support because I don’t have a problem. I smoke marijuana (cannabis) for pain, I don’t really see how people get addicted to it, to be honest.”*
(Tom, YP, 16 years).

In addition, young people described other complexities in their life that determined whether they attended the sessions on offer; experiencing poor mental health was a commonly cited reason:


*“I went into a mental health hospital just after you had thingy’d (made a referral) because I went a bit fucked up. So, I couldn’t go”*
(Angelina, YP, 20 years).


*“I had a bad period of my mental health, and I didn’t end up turning up”*
(Dylan, YP, 19 years).

The quotes from Tom, Angelina and Dylan serve to emphasise the vulnerabilities of the population, reinforcing the fact that they could have benefitted from early intervention/low level advice and support.

Following the implementation of MET and SBNT, the drug and alcohol practitioners clearly articulated how the interventions fitted with the characteristics of the service being analysed. The MET approach was perceived by practitioners to align with the skill set that they were already familiar with and self-assured at using. Most practitioners reported positive leadership engagement and felt supported by their managers when trying to deliver the interventions:


*“My manager is very, very supportive of it and is very interested in it as well and so we discussed that, and she says to continue using it because it is very, very effective”*
(Heather, SBNT practitioner [implementation]).

Managers also expressed being supportive of their staff and encouraging them to use the interventions within their usual care work to benefit the young people accessing specialist services:


*“As a group, I suppose Adam (drug and alcohol worker) has experience of SBNT so I don’t think that there is any argument from us that, actually, as an approach, it’s really, really helpful.”*
(Sam, drug and alcohol manager [intervention]).

### 3.3. Outer Setting—Level of User Need and the Importance of Context

Acquiring information on whether the needs of the young person in care are currently being met was a key part of the pilot RCT. An important aspect of this was to reflect on why young people in care use substances. Most professionals clearly articulated the vulnerabilities present for young people accessing the care system:


*“They use them (substances), for lots of different reasons. It could be to block out what’s happened to them so actually they don’t really want to stop it, thank you very much, because it’s the only thing that they can manage with right now. To them it isn’t a problem. It is a strategy I guess to manage their day. Yes, there might be all those risky behaviours, all that stuff going on but it’s the only crutch they’ve got right now. You take that away, they’ve got nothing.”*
(Laura, residential carer).

Professionals also recognised that some young people ‘normalise’ the use of drugs and alcohol and may not identify their own use as problematic as a direct result of being exposed to situations within which substances have been used by family and friends:


*“If they’ve certainly come from families where using substances isn’t necessarily an automatic negative. It’s part of their self-identity and that’s what they do with their friends. That’s how they enjoy their free time. It’s not always something that they consider anything that needs to be worked on.”*
(Cassie, residential carer).


*“Up until now, they don’t see alcohol and drugs (as) problematic. They think it’s a part of life. They’ve seen it with their own families, and that’s part of their living.”*
(Sophie, foster carer).

The acknowledgement that this group of young people have multiple vulnerabilities and exposure to risky behaviours suggests additional need for interventions to counter the understanding that risky substance use is an inevitable reality for young people in care. Additionally, the above quotes highlight the precise reasoning as to why MET may be a useful intervention with this population. For young people who are not treatment/intervention ready, the ethos of the MET intervention aligns with this and can help people identify and develop motivation to change behaviour.

Once the study started, the wider contextual factors impacting on the ability of services to screen children for substance use and deliver early interventions became obvious. The impact of OFSTED (UK Office for Standards in Education, Children’s Services and Skills) inspections, re-tendering of services, and teams functioning below required staff capacity highlighted the pressures to which services were exposed. Within the duration of the pilot RCT, three out of the six local authority teams reported that they were operating with open vacancies and managers on long term sick leave which impacted on leadership:


*“I’m not making excuses for the service, but we have had different managers off sick. So, somebody who would have been key and pivotal to doing that, wasn’t here for a number of months; there’s another assistant manager been off, nearly six months would have carried that vacancy. So, for them, this [the implementation of SOLID] has just been, I guess not as high priority as it should have been.”*
(Pat, Social work manager [implementation]).

OFSTED inspect services providing education and skills for learners of all ages. Two sites reported preparing for and participating in OFSTED inspections; this resulted in ‘core business’ being prioritised, and all other commitments being suspended. There was no clear definition of ‘core business’ but in the context of social care to support looked after children this was assumed to be dealing with issues that presented immediate risk and fulfilled statutory duties. There was an acknowledgement that taking an early intervention approach was just another task to be undertaken ‘on top’ of the other commitments that teams had:


*“It was another ask on top of all the other asks…. it did come in at a difficult time for the service, both in terms of OFSTED and numbers of staff we have in post.”*
(Ken, social work manager [implementation]).

Four out of six of the drug and alcohol services went through a contract re-tendering process during the study period. Retendering meant that existing service providers had to submit a formal proposal and associated costs to commissioners to continue delivering the service. One practitioner described feeling personally ‘empowered’ following the study training, however the reality of changes in line-management, commissioned service provider and building location meant that organisational barriers affected the ability for some practitioners to stay involved as retendering brought additional pressures to realign with the expectations of a new service contract:


*“I came back to the workplace and, specifically for me, with the change of two buildings, that’s come in due to being TUPED* over to a new service, it was kind of just forgotten about, if I’m completely honest.”*
(Georgia, MET practitioner [implementation]).


** TUPE—Transfer of undertakings (protection of employment) = when an organisation or service transfers to a new employer.*


### 3.4. Characteristics of Individuals—Professionals’ Knowledge, Beliefs and Self-Efficacy Regarding Interventions

In relation to the knowledge and beliefs about the interventions, most of the practitioners were enthusiastic regarding the expected benefits of delivering the interventions and had considered how the implementation of MET/SBNT could enhance their practice and bring ‘added value’ to the interventions they provided to young people in care.

Training was delivered in both the SBNT and MET approach to increase levels of self-efficacy and to ensure that individuals were equipped to execute the necessary courses of action to achieve implementation goals. The SBNT training provided practitioners with knowledge of different techniques and methods of engaging young people within sessions. Practitioners portrayed a belief that:


*“The concept of it all, again, is really good. You can’t deny, on paper, it’s good, it really is… Because everything’s in place for it to be really good, a successful thing.”*
(Karl, SBNT practitioner [intervention]).

The MET practitioners generally expressed a belief in their own capabilities to deliver the adapted interventions:


*“I’m confident with it and I can see that in the progression of how my conversations go with the clients…. I’m using it for my own tier 3 clients with success.”*
(Amy, MET practitioner [intervention]).

Interestingly, despite MET being described as closely aligned with Motivational interviewing training (which all practitioners had received prior to the study) some practitioners felt that training in MET had increased their competence but decreased their confidence levels of delivering the intervention; i.e., it highlighted gaps and anomalies in their knowledge and practice. One participant stated:


*“I am relatively confident when it comes to engaging with clients. I think, doing the training made me less confident, because it highlighted all the stuff … the things I am not doing.”*
(Frank, MET practitioner [intervention]).

### 3.5. The Process—The Implementation and Use of Interventions

The level of engagement in training and supervision from drug and alcohol practitioners, was variable. They were allocated to receive training in and deliver either MET or SBNT. Practitioners allocated to MET or SBNT received two full days’ training in the adapted allocated intervention. Training for each intervention took place at a specialist addiction service and was facilitated by two experienced clinical members of the research team. The feasibility of intervention delivery also took into consideration the responses to the MET and SBNT training and supervision of the practitioners. One of the trainers stated:


*“By and large, these people came along with positive attitudes to training….as a trainer I want to motivate these people to practice in a particular way, because it’s good for them. It’s good for their practice and it’s good for their CVs, and it’s good professional experience for them. I think, in the main, they were very open to it.”*
(Paula, Clinical trainer/supervisor [implementation/intervention]).

Alongside the training, practitioners were offered monthly clinical supervision for the duration of the study. However, the consistency with which practitioners engaged with the clinical supervision was variable:


*“It was very mixed. There were a few highlights of people really doing hands on work, and that was good. Then that went to another level where people aren’t doing it. But there were a handful of therapists that really ran with the idea and did really decent work.”*
(Dale, Clinical trainer/supervisor [implementation]).

Commitment to clinical supervision was difficult to gauge as some drug and alcohol practitioners elected not to attend due to not actively delivering interventions to young people in care as part of the study trial. They therefore deemed the clinical supervision sessions to be unnecessary. However, it should be noted that practitioners could have used the interventions with any clients and taken part in supervision.

Professional participants clearly articulated that the number of steps in the care pathway and the potential for a time lapse to occur between the social worker completing the screening with a young person and the eventual step of a referral onto an external drug and alcohol service, was a barrier to retaining young people within the interventions:


*“Particularly with young people and again the looked after children that momentum has to be jumped on quite quickly. I think there has been a bit of a time lapse from where the referral has been completed and the researchers have been able to visit the young person to when it comes to us…you have to be shit hot with any young person because a day can be a lifetime. Four weeks could be like a previous life.”*
(Olivia, SBNT practitioner [implementation]).

There was consensus between the professionals that once a client had been identified, the first session of a drug and alcohol intervention needed to happen within a week to engage a child in care:


*“I think it would have to be really trying to at least be within that same week, if at all possible. I know that is quite a high and very quick turnover, but I do think it is important to get in there very quick.”*
(Laura, Residential carer [implementation]).

Finally, when practitioners reflected on their usual practice, they described often working in an unplanned way, that did not follow a structured approach. Practitioners reported waiting to see what the young person presented within the session and then using their professional knowledge and skills to facilitate the session:


*“I prefer to wing it a little bit to see how conversations develop.”*
(Christopher, SBNT practitioner [implementation]).

When practitioners considered using a more formal and manualised approach, which was a pre-requisite of the structured and time limited MET and SBNT interventions, they typically reported that they perceived it led to them being more focused and productive during sessions with young people:


*“In our work it is easy just to go and link up for an hour, and have a bit of a natter, and stuff. Whereas, it is a lot more productive (using a structured approach), I think, isn’t it?”*
(Frank, MET practitioner [intervention]).


*“I use the stuff from the training…because it helps me to stay focused and not dilly dally off down some other road.”*
(Amy, MET practitioner [intervention]).

Despite practitioners acknowledging the potential of the interventions as identified above, they also identified that, within their personal practice, they would utilise multiple techniques within each session. They implied that going forward they felt they would deliver elements of the SBNT or MET interventions alongside other approaches such as Motivational Interviewing, Cognitive Behavioural Therapy and counselling techniques:


*“I would definitely use it how I use any other intervention, as and when and if it was appropriate.”*
(Emma, MET practitioner [intervention]).

One of the main barriers of using either the MET or SBNT approaches as perceived by the therapists was that, in their current form, they did not allow the therapeutic relationship to be established before ‘work’ commenced:


*“I know, myself, that I’ve sometimes had two or three sessions, before I’ve even been able to do an assessment. Because, of that firefighting, because of that building the relationship with kids who are just not interested. Trying to motivate them to get involved. By that point with SBNT, you’d have done half your intervention.”*
(Dianne, SBNT Practitioner [implementation/intervention]).

Practitioners redefined how they believed interventions would be used, stating that they would add the new skills into their ‘toolbox’ of techniques and use their professional integrity to establish when it was appropriate for the skill sets to be used.

## 4. Discussion

Findings from this implementation evaluation of the SOLID trial highlight the difficulties of delivering a screen and referral intervention for problem drug and alcohol use in looked after children in a social work and drug and alcohol intervention system that is stretched to capacity. In line with previous studies [25,26] our work suggests that the understanding of what constitutes risky substance use may contrast between different participants. The risk of drug and alcohol use can become normalised both for the young person and the social care system that is set up to protect them. If a young person has experienced repeated exposure to parental or familial substance use prior to being in care and substance use has been normalised within a young person’s lifestyle, this may impact their awareness of risk, their personal immunity to risk, their desire or need to change and therefore their willingness to accept support from specialist services [27]. Despite reporting substance use within the preceding 12-month period, only 15 of the 76 (20%) young people allocated to MET or SBNT attended any of the intervention sessions [23]. This, in turn, became a common barrier to implementation for most drug and alcohol workers as it restricted the opportunity to practice the approaches. However, for some workers, the approaches were making an observable difference to their practice.

This study highlights that the looked after population do need to be supported differently from the general population. The MET approach was described as complementing current practice and skills so it could potentially be used in practice to enhance a client’s motivation and address needs which have previously been given little priority [21]. Young people in care are identified as having multiple vulnerabilities, including experiencing severe mental health problems. Despite substance use being indicated in our sample, many young people reported to be more concerned about other problems in their lives and believed that substance use was a symptom not a cause of problems [23]. Consequently, future work needs to begin with the priorities of young people in care and any work on substance use needs to be set into a wider context of other mental health impacts [28], and difficulties experienced in care [29].

When considering the drug and alcohol environment and the mixed responses to the introduction of SBNT and MET, the findings suggest an increased focus on participants readiness to change is necessary [30]. Despite the principles of the approaches fitting with drug and alcohol practitioners’ values, there were tensions between the new practice of delivering time limited and focused sessions and the wider culture of young people’s drug and alcohol services. Participants highlighted the challenges of trying to use therapeutic approaches with young people in a prevention-focused (earlier intervention) framework as opposed to the traditional problem-focused drug and alcohol practice. This is of relevance to implementation efforts within voluntary and 3rd sector organisations directed by the contractual requirements imposed by commissioners which are often treatment focused [31]. Data within this study highlights that the available system must be responsive enough to intervene efficiently in a time sensitive manner and this may require additional resources than the current referral system affords [23]. However, securing additional resources within the public sector during times of austerity could create barriers to meaningful change [32].

Professional participants possessed a noticeable wealth of skills and knowledge developed through years of clinical practice. Therefore, although current practitioners were trained in MET or SBNT interventions, within the lifetime of this study, half of the drug and alcohol services were re-tendered. The re-tendering process resulted in disruption to service priorities, affected the ability for some practitioners to remain involved in the study and led to interruptions regarding the delivery of the adapted interventions. This level of organisational change and associated staff turnover resulted in a lack of continuity of trained practitioners not only with the expertise to provide the SBNT or MET service offer but also with vulnerable young people. New cohorts of staff had to be trained. This highlights the requirement for services to embed a robust training plan, to ensure that new staff can obtain the requisite skills [23].

The characteristics of individual practitioners involved in the research process and the delivery of new interventions including attitudes towards the new ways of working, and the levels of confidence regarding interventions, also influenced the implementation process. Existing implementation science literature suggests that it takes between two and three years for a social care intervention to reach full implementation, and that it is likely to happen in stages [33]. The findings within this study highlight the diverse range of actors involved in implementing the SBNT or MET behavior change interventions. Each drug and alcohol agency had a particular hierarchy and structure made up of practitioners and managers [34]. The expectations regarding the study and levels of buy-in differed between sites and the drug and alcohol services interacted in varied ways with the social care teams. As a direct result of this, drug and alcohol agencies did not all function in the same way and successful intervention delivery was dependent on multiple interactions within the drug and alcohol system, such as peer support, support between management and frontline practitioners and the policy content in terms of tendering.

Organisational change literature reports that where an organisational cultural shift is being attempted, continuous reinforcement is necessary if the change is to be sustained [30]. This is something we found to be a challenge, especially as the SBNT and MET interventions were being implemented as part of a pilot RCT; as such, the longevity of the implementation effort was uncertain. In addition, organisational factors meant that day to day priorities took precedence. Professionals reverted to old familiar behaviours such as using standard referral criteria to determine the appropriateness of intervention delivery rather than offering interventions in a proactive, risk reduction manner as intended within this study [23].

The context in which the research process occurred, and the interventions were delivered, were complex and involved interdependent interactions within and across statutory social workers and drug and alcohol provider organisations [34]. The interventions used within SOLID were adapted using co-production processes and techniques with multiple different stakeholders involved in the study [21]. This was significant as many clinicians have increased confidence in an intervention if they are developed within their local context [35].

At the time of this trial, health and social care services were under significant pressure. The trial was conducted in 2016–2017 amid prolonged UK austerity measures following the global financial crisis. During this time, public services were being cut and under increasing pressure due to increasing demand [36]. In this context, it is unsurprising that there was an ‘inability’ from services to engage with novel interventions due to other challenges upon them. Prior to this study, knowledge regarding the levels of need surrounding substance use within the looked after population was limited. This study provided local authorities with data from 860 young people, of which 211 (24.5%) indicated they had used substances within the last 12 months and therefore screened positive for being at risk of substance misuse. The data helped to provide more clarity regarding the size and extent of the problems within different localities [23].

## 5. Strengths and Weaknesses of the Study

It was important throughout the study that practitioners had the necessary time for ongoing reflection to occur. The time taken to adapt the interventions, provision of training, ongoing communication with the research team and monthly provision of clinical supervision attempted to provide this necessary opportunity to reflect and evaluate. The large qualitative data set, triangulation of results and theoretically framed analysis are additional strengths. The real-life struggles of a care service have been evaluated but the difficult social and economic environment that the services faced should be noted. The limitations of the study include the fact that the implementation evaluation was undertaken as part of a trial rather than part of usual care. This could have affected buy in to the intervention procedures. In addition, the study was situated in one geographical area of the UK at a particularly difficult time of national funding cuts, limiting generalisability. However, our relatively large qualitative sample, prolonged engagement through time and the fact we were able to engage with 6 local authority sites to some extent mitigated these weaknesses.

This article demonstrates how we have generated actionable findings that would provide future research studies working with young people in care, social work environments and drug and alcohol services with information about the context that may have influenced the implementation process. Environments that are increasingly complex such as social care within which individuals have multiple competing needs and workers are overstretched and under resourced. Unfortunately, this means it is more difficult to implement and sustain change to systems [37,38] without additional resources being allocated. These findings can be used to inform future researchers and stakeholders on potential changes that need to occur to improve the implementation process.

## 6. Recommendations for a Future Evaluation

The screening method used, and the young people’s risk perception levels were problematic. As identified, the validated tools used did detect risky drug and/or alcohol use, however, the uptake of services was low. If the same screen and treat model was used in a future study, a researcher embedded within the children’s social care team could complete the screening. An embedded researcher would have the skills and capacity to sensitively explore the impact of an individual’s drug and/or alcohol use on their physical and mental health, their behaviour and their relationships. There would be an opportunity to deliver an opportunistic intervention with a problem feedback component which in turn may have initiated a stronger recognition of risk on behalf of the young person.

As described, the study tried to deliver the adapted interventions using pre-existing drug and alcohol referral pathways. An alternative way of working would be for a drug and alcohol practitioner to be embedded within the children’s social care team. This would align with the principles of place-based approaches that argue that the providers of services (drug and alcohol practitioners and social workers) work collaboratively to improve the health and care for the populations they service (young people in care). The embedded nature of both the researcher and drug and alcohol practitioners within the social care system would help to improve the engagement of social workers and provide an increasingly personalized response depending on information provided on behalf of the young person [22].

An alternative approach not tested in this study but supported in the wider literature would be to deliver a universal intervention delivered to all young people in care regardless of their substance use history. This approach however would require education and awareness raising training to be delivered to professionals involved in supporting young people in care such as social workers, foster carers, residential workers and drug and alcohol practitioners. An educational approach would require professionals to stop normalising use in this population and instead focus on the cumulative risk for young people in care, not viewing their drug and/or alcohol use in a silo but considering it in relation to the other adversities they have been exposed to.

## 7. Conclusions

The interaction between individuals, service level dynamics and external policy related factors influenced the implementation of new interventions and new ways of working to respond to the needs of children in care. It is essential that the ‘preparing for change’ phase of any new intervention implementation is managed effectively. Novel interventions need to dovetail into the often-complex lives of young people in care and align with the drug and alcohol practitioners’ and social workers priorities to be successfully implemented and become part of routine practice.

## Figures and Tables

**Figure 1 ijerph-18-03461-f001:**
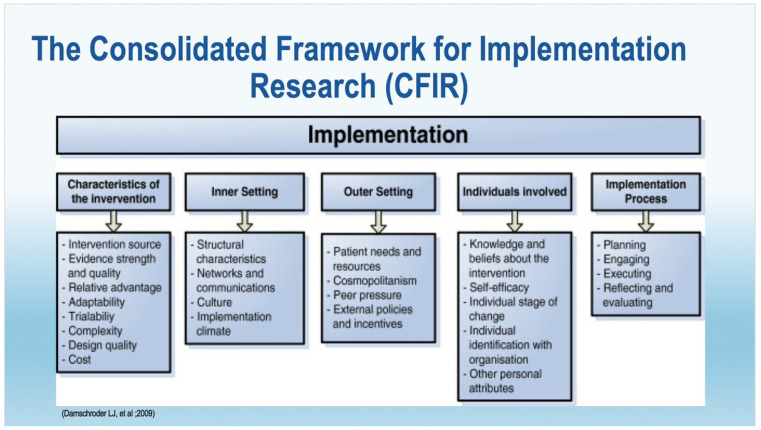
The Consolidated Framework for Implementation Research (CFIR).

**Table 1 ijerph-18-03461-t001:** Participant characteristics.

Qualitative Method	Participant Group	Number of Participants	Gender
1:1 interview	Children in care	37	23 Female; 14 Male
1:1 interview	Drug and alcohol practitioners Drug and alcohol service managers	13 3	10 Female; 3 Male 2 Female; 1 Male
1:1 interview	6 Residential carers; 1 foster carer	7	6 Female; 1 Male
1:1 Interview	1 Social worker strategic manager; 6 Senior social workers/Team leaders; 3 Personal advisors within social work teams; 1 Social work student	11	6 Female; 5 Male
Focus group	Drug and alcohol practitioners	3	2 Female; 1 Male
Focus group	Residential workers	9	4 Female; 5 Male
Focus group	Foster carers	6	4 Female; 2 Male
Focus group	Residential workers	4	1 Female; 3 Male
Focus group	Residential workers	4	2 Female; 2 Male
Focus group	1 social worker; 2 senior social workers/team leads	3	2 Female; 1 Male
Focus group	3 social workers; 2 Personal Advisor	5	5 Female
Focus group	2 social workers; 1 senior social worker/team lead; 1 Personal Advisor	4	4 Female
Focus group	3 Social workers; 1 senior social worker/team lead	4	2 Female; 2 Male
Total	7 participants interviewed twice (2 drug and alcohol practitioners; 3 personal advisors, 1 social worker and 1 senior social worker/team lead) *	113 participants	

* Participants interviewed twice, was to capture data during ongoing recruitment and intervention delivery in July–October 2017 and to capture reflections from practitioners once the randomised controlled trial had ceased August–November 2018.

## Data Availability

The datasets used and/or analysed during the current study are available from the corresponding author on reasonable request.

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
