# Peer review of "Implementation Factors for the Delivery of Alcohol and Drug Interventions to Children in Care: Qualitative Findings from the SOLID Feasibility Trial"

_ijerph, 2021, doi:10.3390/ijerph18073461_

Round 1
Reviewer 1 Report
The offer of effective preventive interventions around alcohol and drug misuse to young people in care is an interesting and potentially important topic. Two apparently contrasting methods and a control group were included in this pilot study and this could have provided valuable evidence about the effectiveness of the interventions. However, two different kinds of issues limit the value of this paper.
Apparently only a tiny number (15) of the screened-in participants actually attended any of the intervention sessions. Two kinds of explanations are offered for this, the first is that the intervention was not seen as very relevant by many of the young people and perhaps also the professionals working with them. The second is that the circumstances in which the drug and alcohol treatment teams were working were undermined by organisational factors, notably the consequences of austerity policies. There is some striking evidence about the positioning of these services.
However, the other problem with the paper is that significant information is not provided or is unclear.
- In the discussion (and this should be reported as a finding) it is stated that only 15 of 76 young people screened in for one of the experimental arms of the study attended intervention sessions. Earlier the paper reports that 37 young people were interviewed. We are not told how many were allocated to the control. The reader needs clear information about the numbers of young people screened, screened in and allocated to each of the three arms of the study. We need to know more about those young people, minimally their age, ethnicity and reasons for being in care. Some measures of drug and alcohol use would also be necessary: which substances, at what frequency, at what level? Did the three groups have similar characteristics? If the time between screening and the start of intervention was important, what were those times in this pilot?
- Similarly there are very few details about the professionals involved, such as their professional roles, their background and qualifications.
- No details are given about the interventions actually provided under each study arm. What evidence is there that professionals trained in MET were actually working differently from those using SBNT or their usual practice?
It is always difficult to write up a study which has not been executed as intended, although the difficulties encountered in failed trials can be very informative. However, sufficient information needs to be given about the study for readers to be able to form a judgement about the lessons to be learnt and I fear that, at the moment, this is not the case with this paper. If, as the title suggests, the intention is for the paper to focus only on problems of implementation, this should be apparent in the introduction which, at present implies the paper is going to report on a successfully executed trial. Where is the discussion of theoretical and empirical evidence of factors which lead to implementation difficulties? More detailed evidence about the difficulties encountered and the stage of the process that they occurred - at the point of access, assessment, randomisation, intervention etc. - should be given.
Author Response
|
Reviewer 1 |
|
|
The offer of effective preventive interventions around alcohol and drug misuse to young people in care is an interesting and potentially important topic. Two apparently contrasting methods and a control group were included in this pilot study and this could have provided valuable evidence about the effectiveness of the interventions. However, two different kinds of issues limit the value of this paper. Apparently only a tiny number (15) of the screened-in participants actually attended any of the intervention sessions. Two kinds of explanations are offered for this, the first is that the intervention was not seen as very relevant by many of the young people and perhaps also the professionals working with them. The second is that the circumstances in which the drug and alcohol treatment teams were working were undermined by organisational factors, notably the consequences of austerity policies. There is some striking evidence about the positioning of these services.
|
Thank you for highlighting the positive contributions of this paper. The important comment about small numbers of children that attended the clinical sessions is well made. We have expanded our introduction to highlight this and cross referenced to our detailed RCT paper and final NIHR report. See lines 85-91. This paper specifically wanted to explore the qualitative findings from our implementation evaluation in detail. As such we have cross referenced to our other papers but maintained the clear qualitative focus of this paper. |
|
However, the other problem with the paper is that significant information is not provided or is unclear.
|
As noted above the current paper highlights the qualitative findings from our implementation evaluation. A summary of the RCT figures and a consort diagram have been included for context. A reference to the published RCT paper and final report has been highlighted. Lines 85-91. |
|
We agree, information regarding D&A practitioners training has been added on lines 120-123. |
|
Details of the MET intervention and the appropriate references are provided on line 57-65. Details of the SBNT intervention and the appropriate references are provided on line 66- 76. The extent to which MET/SBNT was delivered as per the training is difficult to interpret due to the limited number of children seen and the lack of engagement with supervision. Additional information acknowledging this as a limitation is added in lines 128-134. |
|
It is always difficult to write up a study which has not been executed as intended, although the difficulties encountered in failed trials can be very informative. However, sufficient information needs to be given about the study for readers to be able to form a judgement about the lessons to be learnt and I fear that, at the moment, this is not the case with this paper. If, as the title suggests, the intention is for the paper to focus only on problems of implementation, this should be apparent in the introduction which, at present implies the paper is going to report on a successfully executed trial. Where is the discussion of theoretical and empirical evidence of factors which lead to implementation difficulties? More detailed evidence about the difficulties encountered and the stage of the process that they occurred - at the point of access, assessment, randomisation, intervention etc. - should be given.
|
Further details regarding it not being feasible to progress to a full trial has been added and previously published. These papers providing additional data are referenced accordingly in lines 85- 91. |
Reviewer 2 Report
The manuscript by Hayley Alderson et al., titled "Factors That Promote or Inhibit Implementation of Manual Guided Alcohol and Drug Psychosocial Intervention Delivery to Children in Care: Qualitative Findings from a Feasibility Trial", has some interesting concepts and observations about young addiction predisposition in young people in care. The authors study the results by applying Motivational Enhancement Therapy and Social Behaviour and Network Therapy using interviews and focus groups to explore the key lessons from implementing the interventions. In conclusion, AA.s affirm that the implementation need interventions "are delivered in a way that can accommodate the often complex lives of young people in care and align with the drug and alcohol practitioners’ and social workers priorities". I think that these conclusions need to be confirmed by some data and statistical analysis.
Author Response
|
Reviewer 2 |
|
|
The manuscript by Hayley Alderson et al., titled "Factors That Promote or Inhibit Implementation of Manual Guided Alcohol and Drug Psychosocial Intervention Delivery to Children in Care: Qualitative Findings from a Feasibility Trial", has some interesting concepts and observations about young addiction predisposition in young people in care. The authors study the results by applying Motivational Enhancement Therapy and Social Behaviour and Network Therapy using interviews and focus groups to explore the key lessons from implementing the interventions. In conclusion, AA.s affirm that the implementation need interventions "are delivered in a way that can accommodate the often complex lives of young people in care and align with the drug and alcohol practitioners’ and social workers priorities". I think that these conclusions need to be confirmed by some data and statistical analysis. |
The data that supports the statement made in the conclusion is supported in lines · 173- 193- the current drug and alcohol perspective focused on treatment rather than early intervention · 202- 210 discussing the complexities in the YP lives that prevented them from attending (MH) · Section 3.3 discussing the importance of context, specifically lines 254- 289 within which OFSTED inspections took over the ability to screen for substance use as planned and re-tendering influenced the ability to fully engage in delivering the interventions. · Statistical analysis regarding the feasibility of the RCT and the MET/SBNT interventions is provided in an accompanying publication and is referenced within this manuscript- lines 85-91. |
Reviewer 3 Report
This paper examines implementation of the MET and SBNT interventions in the context of a comparative effectiveness RCT focusing on children in care.
The careful focus on implementation is a strength, as is the use of the Consolidated Framework of Implementation Research.
Methods are solid. Qualitative data was collected from an impressive range and number of participants.
Findings and discussion are interesting and generally useful. However, while the discussion does a good job of summarizing and contextualizing challenges, it is a bit light when it comes to recommendations and potential solutions. I would like to have seen this aspect developed a bit more, to maximize the usefulness of the article.
The manuscript could benefit from a careful proofreading and copy editing. For example:
-
Abstract: We concluded that unless interventions are delivered in a way that can accommodate the often complex lives of young people in care and align with the drug and alcohol practitioners’ and social workers priorities, it is [SHOULD READ "THEY ARE"] unlikely to be successfully implemented and become part of routine practice.
- Pg. 2 Line 73: “careers” is misspelled.
- Informed written consent was obtained prior to each interview or focus group took place [perhaps fix this by changing to "taking place"]
These are a few examples, not meant to be a comprehensive list.
Author Response
|
Reviewer 3 |
|
|
This paper examines implementation of the MET and SBNT interventions in the context of a comparative effectiveness RCT focusing on children in care. The careful focus on implementation is a strength, as is the use of the Consolidated Framework of Implementation Research. Methods are solid. Qualitative data was collected from an impressive range and number of participants. Findings and discussion are interesting and generally useful. However, while the discussion does a good job of summarizing and contextualizing challenges, it is a bit light when it comes to recommendations and potential solutions. I would like to have seen this aspect developed a bit more, to maximize the usefulness of the article.
|
Thank you for your comments. I have added in details for recommendations for a future evaluation- lines 510- 539 |
|
The manuscript could benefit from a careful proofreading and copy editing.
|
The manuscript has been proofread again. |
Reviewer 4 Report
The paper “Factors That Promote or Inhibit Implementation of Manual Guided Alcohol and Drug Psychosocial Intervention Delivery to Children in Care: Qualitative Findings from a Feasibility Trial” describes an interesting study that investigates the potential barriers and facilitators to successfully implement two interventions for children in care. The paper addresses an important topic with relevant practical implications, and uses appropriate methods and analyses. There is much to like in this paper; however, it can be improved on some aspects.
Method:
- Why did you not also do focus groups with young people (instead of only interviews)? I actually expected that and thought that would make a lot of sense, since you can also see the group dynamics there (relevant for alcohol use among young people).
- You only select young people who had indicated to drink. But you also said that preventing was the goal of the interventions. Maybe you can elaborate a bit more why you did not include all young people in care.
- Furthermore, related to this, in the results you wrote that young people did not recognize the label of “being a problematic drinker/having a problem” and did not show up for the intervention. If you would not have used this as a selection for the intervention, maybe this problem of no-show and resistance to the label would have disappeared. Perhaps the authors can reflect on this possibility?
Results
- Perhaps you could structure the results a bit more according to the group who you are discussing. I especially find the distinction between the professionals and young people interesting. Maybe you can use headers (within your current structure) emphasizing when you talk about which group?
- I see some quotes that are really about the implementation of the intervention and some about the intervention itself (e.g., the quote by Dianne on p9). Maybe you can make a distinction between these types of statements, because I believe that adjusting the intervention takes a different approach than changing how it is implemented.
Discussion
- I would start the discussion with a stronger focus on what you found (i.e., barriers/facilitators of implementation of these two interventions), and try to stay close to that. Some statements (e.g., p9, line 381-386) are a bit speculative based on your interview dat. E.g., on p9 you cannot be entirely sure that the children have lower risk perceptions (to dot that I think you need a more quantitative study focusing on relations between concepts). I advise the authors to make such claims with a bit more caution.
- I am curious whether you have any data on how well the professionals executed the interventions according to the intervention protocols. And can you say anything about the effects of these interventions, perhaps depending on how well they were implemented/executed? Or will that be part of another paper?
Minor comments
- the title is too long. Maybe you can just mention “two interventions” (or something like that) instead of “Manual ….intervention”
- abstract, line 22. Should it not be aged, instead of age?
- abstract “young people … control”, this sentence is a bit long/complex. Maybe change into two more simple sentences
- 43; care leavers. Are those people who leave (are finished with) care? maybe use that? I was a bit confused.
- You use a lot of abbreviations. I get that, but when possible I would try to avoid this (e.g., looking at p2, it’s full of abbreviations, which does not increase reading flow).
- Table 1: I would change the order of number and gender, so instead of Female 13, I would use 13 females.
- p11, line 473. Should it not be “meaning”?
Author Response
|
Reviewer 4 |
|
|
The paper “Factors That Promote or Inhibit Implementation of Manual Guided Alcohol and Drug Psychosocial Intervention Delivery to Children in Care: Qualitative Findings from a Feasibility Trial” describes an interesting study that investigates the potential barriers and facilitators to successfully implement two interventions for children in care. The paper addresses an important topic with relevant practical implications and uses appropriate methods and analyses. There is much to like in this paper; however, it can be improved on some aspects.
Method: · Why did you not also do focus groups with young people (instead of only interviews)? I actually expected that and thought that would make a lot of sense, since you can also see the group dynamics there (relevant for alcohol use among young people). |
Thank you for raising this question, I have added further details regarding the methods used and the rationale for using only interviews with young people- lines 102- 108. |
|
· You only select young people who had indicated to drink. But you also said that preventing was the goal of the interventions. Maybe you can elaborate a bit more why you did not include all young people in care.
|
This study focused on secondary prevention, rather than primary prevention. At the outset of the study local authorities wanted to screen and refer children with problem drug and alcohol use. However, they did not have a standardised method to do this. Solid set up a system for screening and providing targeted interventions, in effect we tried to manualise a care pathway within the current system. An explanatory sentence has been added lines 81-82 |
|
· Furthermore, related to this, in the results you wrote that young people did not recognize the label of “being a problematic drinker/having a problem” and did not show up for the intervention. If you would not have used this as a selection for the intervention, maybe this problem of no-show and resistance to the label would have disappeared. Perhaps the authors can reflect on this possibility?
|
I agree with this comment, as identified above the study aimed to provide secondary prevention. However, a recommendation (now added into the paper) is that a universal intervention delivered to all young people in care could potentially minimise the resistance seen. Lines 531-539. |
|
Results · Perhaps you could structure the results a bit more according to the group who you are discussing. I especially find the distinction between the professionals and young people interesting. Maybe you can use headers (within your current structure) emphasizing when you talk about which group?
|
Thank you for this comment. However, we wanted to compare and contrast the understanding and views of different participants in a specific theme. As such the results section has purposefully been written to show the similarities and differences between participant groups. I believe a restructure of these sections would lose that important recognition of differences in opinion/experiences. |
|
· I see some quotes that are really about the implementation of the intervention and some about the intervention itself (e.g., the quote by Dianne on p9). Maybe you can make a distinction between these types of statements, because I believe that adjusting the intervention takes a different approach than changing how it is implemented.
|
This is a really helpful comment I have added a square bracket [Implementation] or [Intervention] to differentiate this- clarity added on lines 162- 164.
|
|
Discussion I would start the discussion with a stronger focus on what you found (i.e., barriers/facilitators of implementation of these two interventions), and try to stay close to that. Some statements (e.g., p9, line 381-386) are a bit speculative based on your interview dat. E.g., on p9 you cannot be entirely sure that the children have lower risk perceptions (to dot that I think you need a more quantitative study focusing on relations between concepts). I advise the authors to make such claims with a bit more caution. |
Thank you for your suggestion. I have restructured the discussion section. |
|
· I am curious whether you have any data on how well the professionals executed the interventions according to the intervention protocols. And can you say anything about the effects of these interventions, perhaps depending on how well they were implemented/executed? Or will that be part of another paper?
|
A paper detailing the supervision and fidelity of the intervention and a final report have been published previously. I have clarified this within the paper- lines 128- 134. |
|
Minor comments · the title is too long. Maybe you can just mention “two interventions” (or something like that) instead of “Manual ….intervention”
|
I have reduced the length of the title as recommended. |
|
abstract, line 22. Should it not be aged, instead of age? |
Changed accordingly |
|
· abstract “young people … control”, this sentence is a bit long/complex. Maybe change into two more simple sentences
|
These sentences have been modified, thank you. |
|
43; care leavers. Are those people who leave (are finished with) care? maybe use that? I was a bit confused. |
Care leavers is a common/preferred social care term; however, I have added clarity in the paper regarding the meaning of the term- line 42-43 |
|
You use a lot of abbreviations. I get that, but when possible, I would try to avoid this (e.g., looking at p2, it’s full of abbreviations, which does not increase reading flow). |
I have taken out unnecessary abbreviations |
|
Table 1: I would change the order of number and gender, so instead of Female 13, I would use 13 females. |
Changed as suggested |
|
p11, line 473. Should it not be “meaning”? |
This sentence has been rewritten for clarity, lines 505-506. |